# Adaptive Wavelet Methods for Earth Systems Modelling

**Nicholas K.-R. Kevlahan** 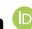

Department of Mathematics and Statistics, McMaster University, Hamilton, ON L8P 2C7, Canada; kevlahan@mcmaster.ca

**Abstract:** This paper reviews how dynamically adaptive wavelet methods can be designed to simulate atmosphere and ocean dynamics in both flat and spherical geometries. We highlight the special features that these models must have in order to be valid for climate modelling applications. These include exact mass conservation and various mimetic properties that ensure the solutions remain physically realistic, even in the under-resolved conditions typical of climate models. Particular attention is paid to the implementation of complex topography in adaptive models. Using WAVETRISK as an example, we explain in detail how to build a semi-realistic global atmosphere or ocean model of interest to the geophysical community. We end with a discussion of the challenges that remain to developing a realistic dynamically adaptive atmosphere or ocean climate models. These include scale-aware subgrid scale parameterizations of physical processes, such as clouds. Although we focus on adaptive wavelet methods, many of the topics we discuss are relevant for adaptive mesh refinement (AMR).

**Keywords:** adaptive mesh refinement; adaptive numerical methods; atmosphere modelling; climate modelling; Earth systems models; large-eddy simulation; ocean modelling; wavelets





## 1. Introduction

This Special Issue of *Fluids* highlights a range of applications of wavelet-based techniques to problems in fluid dynamics, both for data analysis and for the numerical solution of the Navier–Stokes or other fluid equations. In this paper, we review the basic issues and challenges in applying adaptive wavelet methods to solve numerical models in geophysical fluids dynamics. In particular, we focus on the atmosphere and ocean components of Earth system models.

The `matlab` wavelet toolbox has helped make wavelets one of the most popular techniques for analyzing almost any experimental or numerical data set. Interestingly, the continuous wavelet transform actually has its origins in geophysical data analysis: it was originally proposed as a tool to analyze one-dimensional oil well data [1,2]. Since then, wavelet methods have been used to analyze seismic [3,4], ocean [5,6] and atmosphere [7,8] data. They are also a standard tool for image compression [9], denoising [10], and a key ingredient in compressive sampling [11–13]. In general, the continuous wavelet transform is most useful for signal analysis, while the discrete (orthogonal or biorthogonal) transform is used for data compression, denoising and compressive sampling.

In comparison with data analysis applications, wavelet techniques are still much less widely used for the numerical solution of partial differential equations (PDEs). This is true even though the success of wavelets in compressing and analyzing complex multiscale data suggests wavelets are a natural basis for efficiently computing such data! Why should we simulate turbulence using a spectral method, and then post-process it using wavelets? Nevertheless, it has only been in the last 20 years that wavelet methods have been developed and used for the numerical approximation of the underlying dynamical equations. The goal is to use the fact that many natural signals compress well in a wavelet basis to build a dynamically adaptive numerical scheme.

After some initial attempts to develop wavelet Galerkin approaches (to take advantage of both grid compression and operator compression), most wavelet-based methods for PDEs now use the collocation method. In this approach, wavelets are used to provide an adaptive multiscale grid structure (equipped with appropriate prolongation and restriction operators), but discrete differential operators are approximated in physical space using standard finite difference or finite volume methods. Because they are constructed in physical space, wavelet collocation methods use biorthogonal rather than orthogonal wavelets [14].

Adaptive wavelet collocation methods provide a genuinely multiscale alternative to adaptive mesh refinement (AMR) [15–17]. AMR has been used extensively in astrophysical fluid dynamics [18], and to a lesser extent in regional ocean modelling [19–22]. There have also been some initial attempts to apply AMR to global atmosphere models [23].

Adaptive wavelet methods for geophysical flows have been limited primarily to the two-dimensional shallow water equations on the plane [24–26], and small-scale two-dimensional test cases of small scale atmospheric convection [27] and atmospheric boundary layer flow [28]. In addition, a wavelet method has been proposed for three-dimensional atmospheric chemical transport in flat topology [29]. To date, the only global adaptive wavelet models of atmosphere and ocean flow are our WAVETRISK family of models that will be reviewed in Section 4 [30–33]. We will use WAVETRISK development as an example of how adaptive wavelet methods (and AMR) can be applied to Earth systems modelling.

Wavelet methods can be more complex to implement than standard AMR techniques, but they also have advantages. Compared to AMR, wavelet methods do not suffer from wave reflection at refinement boundaries and "hanging nodes" are not an issue. The wavelet approach also directly controls the discretization error and number of refinement levels, using a single non-dimensional tolerance parameter. Mimetic properties (such as mass conservation) can be built into the wavelet transforms. In addition, appropriately designed wavelet methods can provide an adaptive overlay on an existing flux-based method [30,33]. Perhaps the most useful feature of adaptive wavelet methods is that they avoid the need for the ad hoc grid refinement criteria that are inherent to AMR methods [23,33].

AMR and wavelets are both examples of h-refinement: the local grid resolution is modified dynamically, based on certain error indicators. Other techniques to provide dynamically adaptive grid resolution in geophysical simulations include r-refinement (where the number of cells remains constant over time, but the grid is stretched locally to increase resolution) and p-refinement (where the local approximation order is varied).

In addition, nested grids are commonly used in atmosphere models to provide static higher resolution in areas of interest (e.g., higher resolution over land masses [34]). In this method, a local fine resolution grid is embedded in a global coarse grid. Boundary conditions for the fine grid are obtained from the coarse resolution grid, and the fine grid data is restricted periodically to coarse grid simulation. Like AMR, this method can produce reflections at the boundaries of the nested grid, and the procedure for generating the boundary conditions is not well understood.

Variable resolution unstructured triangular meshes, or finite elements, are used extensively in ocean modelling to provide higher resolution near coastlines. Local static refinement has also been proposed to improve representation of orography (e.g., the Andes mountain range [35]). However, this approach can produce instabilities and does not always improve accuracy. A multiscale structure of regular grids may have better numerical properties than a single scale unstructured mesh where the geometrical properties are not easy to control (e.g., triangles can have extremely acute angles). Unstructured mesh generation is an open area of research, especially in aeronautics. On the other hand, the multiscale scale structure of the wavelet method adds significant additional computational overhead (about a factor of two for each active grid point). This means that a grid compression factor of more than two is necessary for the adaptive wavelet method to be more efficient.

However, adaptive multiscale wavelet collocation methods are mostly limited to engineering applications, especially high Reynolds number incompressible and compressible flows. This is surprising since perhaps the most challenging and important applications of computational fluid dynamics are the Earth systems models of the ocean and atmosphere. Adaptivity has the potential to dramatically improve the accuracy and computational efficiency of these geophysical models.

Nevertheless, there are in fact significant obstacles to the use of adaptive wavelet methods, or indeed an AMR approach, in ocean and atmosphere models. These complex models impose specific constraints on numerical methods, and there is therefore a need to implement approximations that have been fully validated by the geophysical modelling community. The goal of this paper is to highlight the properties an adaptive model must possess to be practically useful for atmosphere and ocean modelling, especially as part of a climate model. We illustrate the development of such models in detail using the the WAVETRISK [33] dynamically adaptive wavelet model as an example. WAVETRISK is the first dynamically adaptive global three-dimensional atmospheric model. WAVETRISK represents a first step towards a true dynamically adaptive climate model, incorporating both atmosphere and ocean components.

This paper focuses on the high level problem of how to successfully develop realistic adaptive wavelet methods for atmosphere and ocean models. We will assume a basic understanding of the wavelet multiresolution analysis, and how an dynamically adaptive wavelet method for a PDE is structured. The interested reader could consult [36,37], or other papers in this Special Issue, for an overview of the mathematics and numerical analysis underlying these methods.

In Section 2, we summarize the properties that a dynamically adaptive method must have in order to be usefully applied to atmosphere and ocean climate models. Although we focus on climate models, and especially global models on the sphere, many of these features also apply to regional models on shorter time scales (e.g., numerical weather prediction or coastal flows), as well as to AMR implementations.

Section 3 highlights the specific challenges related to representing the complex topography associated with orography (e.g., mountains), coastlines and bathymetry in geophysical models. Many of these challenges also apply to non-adaptive models, where accurate representation of topography remains a challenge.

Section 4 illustrates how the concepts outlined above have been implemented in actual semi-realistic global atmosphere and ocean models: WAVETRISK and WAVETRISK-OCEAN. The WAVETRISK models provide a practical and successful example of how the challenges described can be met in practice. They also demonstrate that adaptive wavelet collocation models can achieve good parallel performance on standard geophysical test cases.

In addition to these fundamental issues related to numerical modelling of geophysical flows, in Section 5 we briefly summarize two complementary wavelet-based techniques that can be used to improve the performance of adaptive multiscale wavelet-based methods: local time stepping and adaptive multigrid solvers for elliptic equations. Local time stepping adapts the time step to the (local) grid scale. Adaptive multigrid solvers use the adaptive multiscale grid structure produced by the wavelet method as the basis for an elliptic multigrid solver.

Finally, in Section 6 we describe the challenges that remain in developing operational adaptive climate models, focusing on subgrid scale (SGS) modelling and the parameterization of physical processes such as clouds and convection.

## 2. Necessary Properties of Atmosphere and Ocean Models

Adaptive wavelet methods for PDEs have been successfully developed and validated for a wide range of engineering flows [37]. Some examples include incompressible turbulence [38], magneto-hydrodynamics [39], fluid–structure interaction [40–43] and supersonic flows [44]. In most cases the discretization locates all variables at the same grid points, discretization errors are controlled to a finite tolerance and the total simulation times are

relatively short. In addition, the numerical discretizations used do not necessarily have to conform to evolving "state of the art" approximations, since it is relatively clear which discretizations are valid approximations for the particular applications.

The situation is very different for global atmosphere and ocean models. It is vital to understand the particular demands of these Earth systems applications in order to design an adaptive code that will be useful to the geophysical fluid dynamics community. In particular, since the development of numerical approximations is still an active area of research, adaptive wavelet methods should be general enough to be implemented as an adaptive overlay on any flux-based method.

The most basic constraint of geophysical simulations is that they are necessarily extremely under-resolved. Unlike engineering flows, it is not possible, even in simple cases, to resolve all energetically active scales of motion for all physical processes. In the atmosphere and oceans these scales range over at least nine orders of magnitude, from $O(10^6)$ m to $O(10^{-3})$ m. Direct numerical simulations are therefore impossible, and even large eddy simulation (LES) is a challenge. Because geophysical simulations are extremely under-resolved, geophysical fluid dynamics discretizations ensure that certain properties of the continuous PDEs also hold at the discrete level. This is called mimetic approximation. For example, mass must be conserved to machine precision by the discretization. In addition to mass conservation, current models satisfy additional mimetic properties. These may include discrete conservation of energy or enstrophy, conservation of circulation, or exact maintenance of geostrophic balance.

The need for mimetic discretizations has pushed the community to use a variational approach to derive discretizations from the discrete Euler–Lagrange equations, rather than from the dynamical equations themselves [45]. In Section 4, we show how mimetic properties can be conserved in an adaptive wavelet method by appropriate design of the flux restriction and prolongation operators.

Most discretizations for ocean and atmosphere flows are flux-based, and therefore use staggered grids to conserve fluxes. The use of staggered grids complicates the wavelet transform since different variables are located at different grid points or cell edges. Figure 1 shows an example of the so-called C-grid discretization for the shallow water equations. Note that the primal and dual grids have different geometries (triangles and hexagons, respectively), and that each variable is located at a different point. The use of hexagons and triangles is suitable for an icosahedral discretization of the sphere. A wavelet method therefore needs separate scalar valued transforms (on the hexagons) and vector-valued transforms (on the triangles).

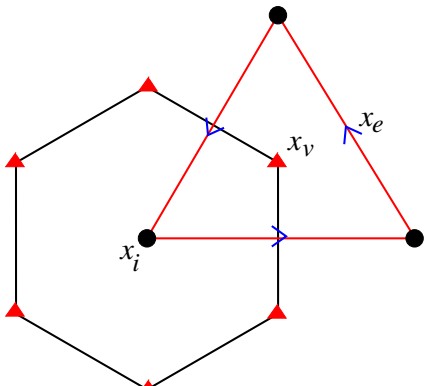

**Figure 1.** The regular hexagonal C-grid. Circulation is located on the primal grid of triangles at points $x_v$. Mass and other scalars such as temperature are located on the dual grid of hexagons at nodes $x_i$. The velocities and fluxes are located on edges $x_e$.

The primal grid is refined by repeated edge bisection of the icosahedron, producing a nested hierarchy of triangular grids. However, on the sphere the dual grids (defined by the edge bisectors of the primal grids) are not nested, and dealing with the overlapping dual

grids is necessary to build the wavelet transform method. An example of how to construct an adaptive wavelet method on a hexagonal C-grid is given in Section 4.

An adaptive wavelet method also needs to ensure that the thresholding of scalar and vector wavelets controls the approximation error of the tendencies. This requires appropriate asymptotic estimates of the PDEs themselves, since the tendencies for each prognostic variable depends on the others through the PDE.

Global models are defined on the sphere. Since the spherical topology cannot be tiled uniformly, the resulting grid is irregular (the geometry of each cell is unique) and there are singular points with different valences. If the coarsest primal grid is the icosahedron, there are twelve pentagonal dual cells in addition to the usual hexagonal cells. Different discretizations of the sphere are possible (latitude–longitude, cubed sphere, ying-yang, . . . ), but every discretization has singular points and irregular geometry. This means that the wavelet transform must take into account local geometrical information and the special singular point structure.

Finally, three-dimensional models are layered, and use either Eulerian or Lagrangian vertical coordinates. Although in principle it would be possible to implement full three-dimensional adaptivity, it is more consistent with the physics and much easier in practice to adapt only in the horizontal direction. This results in a grid which is a set of vertical columns of various sizes. This approach is well-suited to the parameterizations used in geophysical models, which expect to work on columns. One of the most important examples is the vertical diffusion model, which uses an eddy-viscosity type turbulent kinetic (TKE) energy closure. The solar radiation and surfacing forcing models also work on individual columns. Although sacrificing vertical adaptivity limits grid compression, most geophysical flows have strong horizontal gradients.

## 3. Representing Topography in Adaptive Multiscale Geophysical Models

An essential component of atmosphere and ocean models is accurate representation of topography. This is primarily mountains (orography) for atmosphere models and coastlines/bathymetry for ocean models. Topography is naturally multiscale: refining a grid over topography reveals new details at every scale. This multiscale structure is a challenge for conventional models, but is an excellent application for adaptive methods. Adaptive methods make it straightforward to statically refine over mountain ranges or complex bathymetry, so that the associated fluid dynamics is well represented. The challenge is to provide a relatively simple way of refining and coarsening the topography consistent with the adaptive representation of the fluid dynamics. Mathematically, this corresponds to developing a simple multiscale representation of the boundary conditions of the PDE, which may be no-slip (Dirichlet), free-slip (Neumann), or some combination of the two (Robin).

One option would be to fix the resolution of the topography at a particular scale, and simply interpolate or coarsen this base representation as the computational grid adapts to follow the dynamics. While this approach is simple, it loses the multiscale information about the topography that the adaptive approach is able to provide. In addition, choosing a fixed resolution of a coastline, for example, wastes computational resources by over-resolving regions where the flow is smooth.

Consider the example of tsunami propagation. A tsunami is generated seismically in a small region, but then rapidly propagates over very large distances, eventually interacting with coastlines. Wave amplitudes in mid-ocean are very small, but nonlinear steepening and interaction with the details of the coast and shallow bathymetry produces extremely high and damaging waves. The goal of tsunami modelling is to predict the interaction of the tsunami with the coast, and this requires extremely accurate small scale representation of the coastline geometry and bathymetry. So, ideally, one would prefer to have an extremely coarse representation of the coastlines except in those locations where the tsunami has actually reached the coast.

A consistent multiscale representation of the boundary conditions is difficult to achieve explicitly, since the fluid domain actually grows and decreases as the grid resolution

changes. Explicit representation of the boundary conditions also requires a multiscale representation of the one-dimensional geometry of the coastline.

An alternative approach is to enforce boundary conditions implicitly, using a penalization technique. Volume penalization is particularly attractive since the land masses are defined using a (smoothed) mask function: zero in the fluid regions and one in the solid regions. There are no requirements on the smoothness of the actual fluid–solid boundary, and appropriate masks can be generated easily from available databases (e.g., ETOPO [46]).

Volume penalization methods for the Navier–Stokes equations were introduced by [47,48]. This Brinkman penalization is based on the equations for flow through a porous medium. No-slip boundary conditions are approximated by taking the limit of vanishingly small porosity and permeability in the solid regions. Free-slip boundary conditions are approximated by neglecting the permeability (friction) terms [49]. Brinkman penalization enforces the boundary conditions only to first-order accuracy, although this is usually sufficient, especially since h-refinement can be used to reduce the error. A family of higher-order Brinkman penalization methods has been proposed by [50].

Since its introduction, Brinkman penalization has been applied to a wide range of fluid flow problems and numerical schemes, including spectral methods [51], moving boundaries [52,53], the wave Equation [54], the compressible Euler equations [49,55] and the shallow water equations [26,56].

We now review the volume penalization technique proposed in [32] for the shallow water equations. The penalized two-dimensional shallow water equations are

$$\partial_t \tilde{h} + \operatorname{div} \tilde{h}u = 0,$$

$$\partial_t u + \frac{\operatorname{curl} u}{\tilde{h}} \times \tilde{h}u + \operatorname{grad}\left(\frac{g\tilde{\eta}}{\phi(x)} + \frac{1}{2}|u|^2\right) = -\sigma(x)u, \tag{1}$$

where $\tilde{\eta} = \phi(x)\eta$ ($\eta$ is the perturbation of the free surface) and total depth $h(x,t) = H + \eta(x,t)$. The porosity $\phi(x)$ and permeability $\sigma(x)$ are (approximately) discontinuous,

$$(\phi(x), \sigma(x)) = \begin{cases} (\alpha, 1/\epsilon) & \text{in the penalized region,} \\ (1, 0) & \text{in the fluid,} \end{cases} \tag{2}$$

with $\epsilon \ll \alpha \ll 1$, where $\alpha$ and $\epsilon$ are, respectively, the porosity and permeability parameters of the solid (porous) regions. The solid regions are defined by the mask $\mathbb{1}(x)$,

$$\mathbb{1}(x) = \begin{cases} 1 & \text{in the solid,} \\ 0 & \text{in the fluid.} \end{cases} \tag{3}$$

For numerical stability the mask $\mathbb{1}(x)$ is smoothed over a few grid points. The porosity $\phi(x)$ and permeability $\sigma(x)$ are defined based on $\mathbb{1}(x)$ and the control parameters $\alpha \ll 1$ and $\epsilon \ll 1$ as

$$\phi(x) = 1 + \mathbb{1}(x)(\alpha - 1), \tag{4}$$

$$\sigma(x) = \frac{1}{\epsilon}\mathbb{1}(x). \tag{5}$$

This penalization conserves mass and is stable (total energy is decreasing) and has the same wave speed in fluid and solid regions. The error of the penalization is $O(\alpha\epsilon^{1/2})$. Note that only $\epsilon$ constrains the stability of an explicit numerical method, $\Delta t \leq \epsilon$, so the overall accuracy of the penalization can be controlled easily by adjusting the porosity parameter $\alpha$. The constant wave speed and the addition of the porosity parameter $\alpha$ give this method some practical advantages over that proposed by [26].

Figures 2 shows the adapted grid and surface wave height from a penalized shallow water WAVETRISK-OCEAN simulation of the 2004 Indonesian tsunami. Note that the increased resolution is only used where required by the tsunami dynamics (i.e., not at all locations along the coastlines) [32]. The finest resolution is 500 m ($j = 14$ bisections of the

icosahedron) and the coarsest resolution is 16 km ($J_{min} = 9$). Figure 3 illustrates the high sensitivity of the adaptivity to the wavefront dynamics.

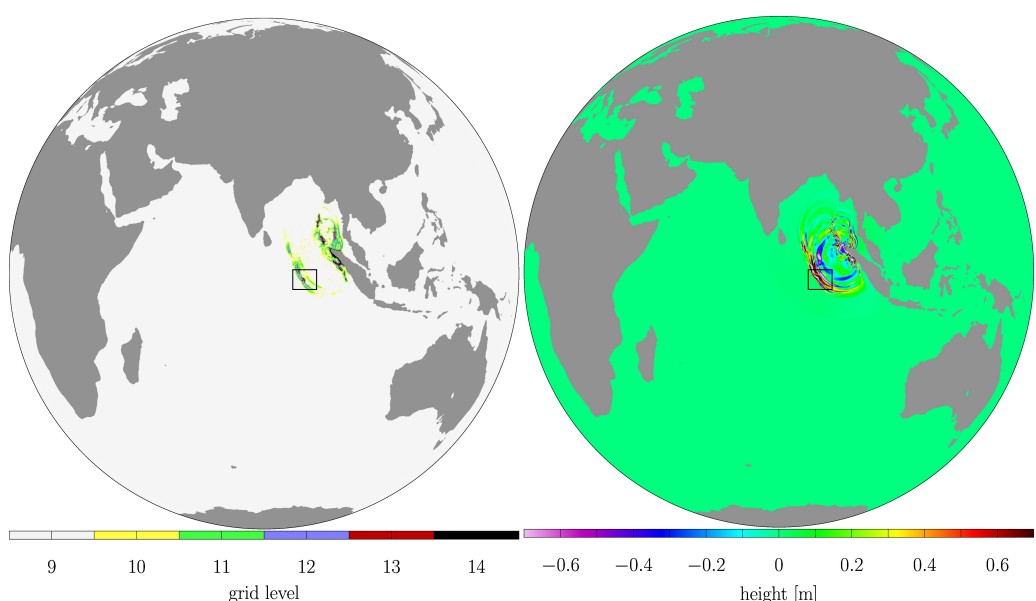

**Figure 2.** Tsunami after 70 min. The grid compression ratio is 930 and the finest resolution of 500 m is required only where the tsunami is interacting with the coastline, and locally in parts of the propagating wavefront. The black boxes indicate the zoomed regions shown in Figure 3. (Reproduced from [32].)

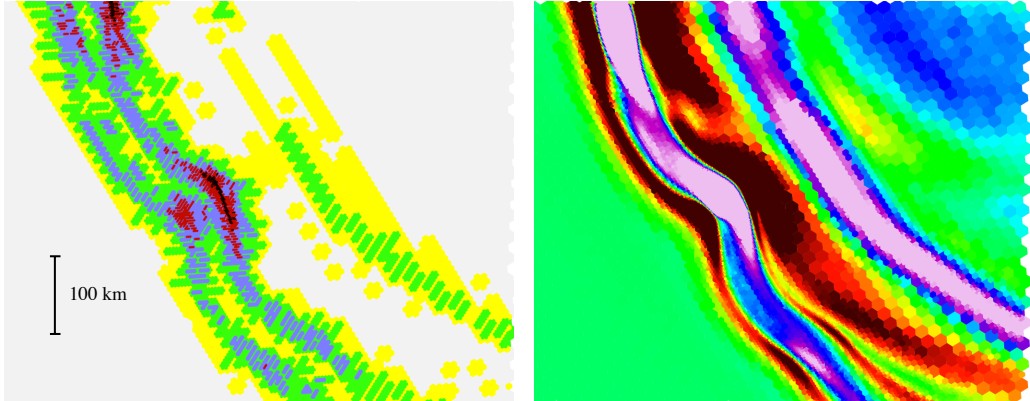

**Figure 3.** Tsunami simulation using volume penalization. approximately 650 km × 550 km zoom of grid (**left**) and height (**right**) for results shown in Figure 2. The black hexagons have size approximately 0.5 km. (Reproduced from [32].)

In recent work [57], we have extended this volume penalization approach from coastlines to bathymetry in three-dimensional ocean models. Volume penalization of small scale bathymetry details provides benefits even in non-adaptive models by avoiding the large pressure gradient errors normally associated with strong bathymetry gradients. We plan to integrate this bathymetry penalization in WAVETRISK-OCEAN.

Although we have focused here on penalization for coastlines and bathymetry in ocean models, an approach similar to that in [57] could be used for orography in atmosphere models.

## 4. The WAVETRISK Global Atmosphere and Ocean Models

### 4.1. Background and Goals

The WAVETRISK project grew out of an initial attempt to develop an adaptive wavelet collocation solver on the sphere [58]. This project used the biorthogonal second generation

wavelet transform on the sphere proposed in [59]. Our intention was to use this two-dimensional code as the basis for a global atmosphere or ocean dynamical core. However, it rapidly became obvious that our engineering approach was not suitable for geophysical flows, because it was not compatible with staggered grids and did not discretely conserve mass. Therefore we started over, incorporating the essential features of a practical geophysical code as described in Section 2.

The basic features of WAVETRISK were introduced in [30] for the shallow water equations on the $\beta$ plane. We used the TRISK discretization scheme [60] on the hexagonal–triangular C-grid shown in Figure 1 because of its excellent mimetic properties and its operational use in the model for prediction across scales (MPAS) code [61]. This scheme is second order in space on the plane. The TRISK scheme conserves either kinetic energy or potential enstrophy. In addition, potential vorticity is conserved, and the Lagrangian behaviour of potential vorticity is consistent with that of the continuous equations (i.e., the material derivative is the same).

The wavelet adaptivity is designed as an overlay on any flux-based discretization. Mimetic properties (e.g., mass conservation) are preserved by the adaptivity, and the discretizations of differential operators are unchanged. The foundations of the method are one-scale operators (in this case, the TRISK discretizations) and two-scale prolongation/restriction operators between a fine scale $j + 1$ and a coarse scale $j$.

In the following section we review the key aspects of the adaptive wavelet method for the shallow water system, focusing on the ingredients needed for a practical atmosphere or ocean model. Full details of the three-dimensional atmospheric model are given in [33] and cited references.

### 4.2. Conservative Adaptive Wavelet Algorithm for the Shallow Water Equations

The two-dimensional shallow water equations for flat bathymetry are

$$\partial_t h + \text{div } F \quad = \quad 0, \tag{6}$$
$$\partial_t u + F^\perp q + \text{grad } B \quad = \quad 0, \tag{7}$$

where $h = H + \eta$ is the total depth ($\eta$ is the free surface perturbation), $F = hu$ is the depth flux, $F^\perp$ is depth flux perpendicular to $F$, $q = (\text{curl } u + f)/h$ is the potential vorticity (with $f$ the Coriolis parameter), and $B = g\eta + K$ is the Bernoulli function (with $g$ the gravitational acceleration and $K = |u|^2/2$ the kinetic energy). (Note that we set density $\rho_0 = 1 \text{ kg/m}^3$ to simplify the equations.) These equations are discretized on the C-grid (see Figure 1) as

$$\partial_t h_i + [\text{div } F_e]_i \quad = \quad 0, \tag{8}$$
$$\partial_t u_e + F_e^\perp \hat{q}_e + [\text{grad } B_i]_e \quad = \quad 0, \tag{9}$$

where $h_i$ is the height at hexagonal node $i$, $u_e$ is the velocity at edge $e$, $F_e$ is the thickness flux normal to a hexagon edge, and $F_e^\perp$ is the thickness flux in the direction normal to a triangle edge. The operators div, grad, curl, and $F_e^\perp q_e$ are discretized using the TRISK scheme [60].

The mass-conserving adaptive wavelet algorithm for the TRISK Equations (8) and (9) is described in Algorithms 1 and 2. Note that we assume the reader is familiar with the basic structure of a second-generation (biorthogonal) adaptive wavelet transform.

The scales are constructed by dyadic refinement from the coarsest resolution $J_{\min}$, so that the primal grid edges satisfy $\Delta x_j = 2\Delta x_{j+1}$. Recall that the scaling functions (e.g., $h_i^j$, no tilde) are the approximations at level $j$, while the corresponding wavelets (e.g., $\tilde{h}_i^j$) give the differences between the scaling functions approximations at levels $j + 1$ and $j$ (i.e., the interpolation error).

An adaptive grid is generated by neglecting those wavelet coefficients (and associated grid points) with magnitude $\leq \tilde{\varepsilon}$. Because TRISK uses a staggered grid, with height and velocity at different points, these algorithms must include additional grid points (i.e., an enlarged index set) to ensure that all values necessary for the computation of the

TRISK operators, as well as the height and velocity wavelet transforms, are present. This represents a significant increased complication compared with a purely collocated method where all variables are located on a single grid!

The key to preserving the mimetic properties of TRISK (and conserving mass) in the adaptive algorithm is to restrict the fluxes, rather than the prognostic variables themselves. Equations (10)–(12) give the constraints these restriction operators must satisfy to preserve the mimetic properties of TRISK.

---

**Algorithm 1:** Conservative adaptive wavelet computation of tendencies.

---

1.  Perform an **inverse wavelet transform** to compute scaling coefficients $h_i^j$ and $u_e^j$ from the wavelet coefficients $\tilde{h}_i^j$, $\tilde{u}_e^j$ and coarsest levels scaling coefficients $h_i^{J_{\min}}$, $u_e^{J_{\min}}$. The data flow is from the coarsest level $J_{\min}$ to the finest level at which non-zero wavelet coefficients exist (determined by the tolerance $\tilde{\varepsilon}$).

2.  Compute **mass flux** $F_e^j$, **Bernoulli function** $B_i^j$ and **potential vorticity flux** $F_e^{\perp}\hat{q}_e^j$ in a loop starting at the finest level:

    (a)  Where possible, compute $B_e^j$, $F_e^j$ by **restriction** from $j+1$.

    (b)  Where restriction is not possible, at level $j$ compute $F_e^j$ and $B_e^j$ using the TRISK operators at level $j$ applied to $h_i^j$ and $u_e^j$.

    (c)  Where possible, compute $F_e^{j\perp}\hat{q}_e^j$ by **restriction** from $j+1$.

    (d)  Where **restriction is not possible**, compute $F_e^{j\perp}$ from $F_e^j$ and $\hat{q}_e^j$ using the TRISK operators at level $j$.

3.  At each level $j$ separately, **apply TRISK operators** to $B_e^j$, $F_e^j$ and $F_e^{j\perp}\hat{q}_e^j$ to compute the scaling function tendencies $\partial_t h_i^j$, $\partial_t u_e^j$, and then obtain wavelet tendencies $\partial_t \tilde{h}_i^j$, $\partial_t \tilde{u}_e^j$ using the **wavelet transform** between levels $j+1$ and $j$.

---

---

**Algorithm 2:** Determining the index sets required for the adaptive wavelet computation of tendencies. This expands the active grid (i.e., produces a larger index set) compared with simply thresholding wavelet coefficients

---

1.  Given the set **active wavelet coefficients** (magnitude $\geq \tilde{\varepsilon}$), determine separately at each level $j$ the **indices required to compute wavelet tendencies** $\partial_t \tilde{h}_i^j$, $\partial_t \tilde{u}_e^j$ from scaling function tendencies $\partial_t h_i^j$, $\partial_t u_e^j$.

2.  Determine which **scaling functions** $F_e^j$ and $B_i^j$ need to be computed, in a loop starting from finest level:

    (a)  Check which $F_e^{j\perp}\hat{q}_e^j$ can be obtained by **velocity restriction** from scale $j+1$. Where this is not possible, determine which scaling function indices **needed to compute the relevant TRISK operators**.

    (b)  Check which $F_e^j$ and $B_i^j$ can be obtained by **restriction** from level $j+1$. Where this is not possible, determine which scaling function indices **needed to compute the relevant TRISK operators**.

3.  Given the set of indices needed to compute the TRISK operators, find the minimal set of indices $i$ and $e$ **needed to compute the inverse wavelet transforms**.

---

To preserve the mimetic properties of the TRISK scheme the restriction operators from scale $j + 1$ to scale $j$ are designed so that they satisfy the commutation properties

$$R_h^j \circ \mathrm{div}^{j+1} \quad = \mathrm{div}^j \circ R_F^j \qquad \textit{conserves mass,} \tag{10}$$

$$\mathrm{curl}^j \circ R_u^j \quad = R_\zeta^j \circ \mathrm{curl}^{j+1} \quad \textit{conserves circulation,} \tag{11}$$

$$\mathrm{grad}^j \circ R_B^j \quad = R_u^j \circ \mathrm{grad}^{j+1} \quad \textit{no spurious vorticity,} \tag{12}$$

where div, grad, curl are the corresponding TRISK operators, $R_h$ is the height restriction, $R_F$ is the flux restriction, $R_u$ is the velocity restriction, $R_\zeta$ is the circulation (vorticity) restriction, and $R_B$ is the Bernoulli function restriction. The third commutation relation ensures that a flow with uniform potential vorticity remains uniform under advection by an arbitrary velocity field (i.e., vorticity is advected like a tracer). Conserving these mimetic properties is especially important for multi-year simulations, where small unphysical effects can accumulate over time.

In addition to the above mimetic properties, TRISK also conserves either kinetic energy or potential enstrophy. When run non-adaptively, WAVETRISK also conserves kinetic energy. However, grid adaptation necessarily involves some dissipation of energy when grid points are removed. Most energy is recovered during interpolation onto the new adaptive grid, but a small amount is lost. Ref. [30] show that grid adaptation is equivalent to Laplacian diffusion with viscosity proportional to $\tilde\varepsilon$. In practice, most climate models include a small amount of diffusion, using a Laplacian or bi-Laplacian operator, to damp grid scale noise and as a crude model of the effect of unresolved scales.

### 4.3. Adaptivity

As outlined in the previous section, the adaptive grid is generated by first thresholding the wavelet coefficients $\tilde h_i^j$ and $\tilde u_e^j$ and then enlarging the required index set by including those points required to compute the TRISK operators and wavelet transforms (i.e., all restriction and prolongation operators). As in other adaptive wavelet methods, the index set is further enlarged by adding nearest neighbours in both position and scale. The nearest neighbours in position are sufficient for a CFL number of one, while the nearest neighbours in scale are sufficient for a PDE with at most quadratic nonlinearities.

Recall that for a single variable $u(x)$, applying a threshold $\tilde\varepsilon$ to its wavelet coefficients $\tilde u_i^j$ controls the wavelet reconstruction error, as well as the number of active grid points $\mathcal{N}$ (and hence the degree of grid compression),

$$\|u(x) - u_\ge(x)\|_\infty \quad = \quad O(\tilde\varepsilon), \tag{13}$$

$$\mathcal{N} \quad = \quad O(\tilde\varepsilon^{-1/2N}), \tag{14}$$

$$\|u(x) - u_\ge(x)\|_\infty \quad = \quad O(\mathcal{N}^{-2N}), \tag{15}$$

where $N$ is the order of the interpolation (prolongation) operator used in the wavelet transform ($N = 2$ for WAVETRISK).

The fact that on the C-grid we have two different wavelet transforms, scalar-valued for $h$ and vector-valued for $u$, on different grids, means that the thresholding operation and subsequent error control is more subtle. We need to define separate thresholds for the wavelet coefficients $\tilde h_i^j$ and $\tilde u_e^j$ that ensure that the errors of the tendencies $\partial_t h_i$ and $\partial_t u_e$ are controlled to the same relative tolerance (note that the tendencies for height and velocity depend on both prognostic variables). We have explored three solutions to this thresholding problem on staggered grids.

In the first case we considered directly thresholding the tendency wavelets. However, this also requires an appropriate thresholding scheme for the wavelet of the variables themselves, and does not give appreciably better results than thresholding just the variable wavelets.

In the second case, for the shallow water equations, we performed an asymptotic analysis on the linearized PDEs to estimate consistent thresholds on the height and velocity wavelets [30]. We derived thresholds in two different regimes: the inertia–gravity wave regime (i.e., fast dynamics, like a tsunami wave) and the quasi-geostrophic regime (i.e., slow dynamics, like two-dimensional turbulence). For the inertia–gravity wave regime we found that to control the tendencies to a relative tolerance of $\tilde{\varepsilon}$ we must apply the following thresholds to height and velocity,

$$\tilde{\varepsilon}_h \sim \frac{cU}{g}\tilde{\varepsilon}^{3/2}, \qquad \tilde{\varepsilon}_u \sim U\tilde{\varepsilon}^{3/2}, \tag{16}$$

where $c = \sqrt{gH}$ is the wave speed and $U$ is a characteristic velocity scale (recall that for simplicity we set the constant density $\rho_0 = 1$ kg/m$^3$, so it does not appear explicitly). A similar analysis for the quasi-geostrophic regime gives the thresholds

$$\tilde{\varepsilon}_h \sim \frac{fULRo}{g}\tilde{\varepsilon}^{3/2}, \qquad \tilde{\varepsilon}_u \sim URo\tilde{\varepsilon}^{3/2}, \tag{17}$$

where $Ro = U/(fL) \ll 1$ is the Rossby number.

Finally, in [33], we showed that a simple thresholding scheme, like

$$\tilde{\varepsilon}_h \sim \|h\|_\infty \tilde{\varepsilon}, \qquad \tilde{\varepsilon}_u \sim \|u\|_\infty \tilde{\varepsilon}, \tag{18}$$

is sufficient to control the tendency errors for linear constant coefficient PDEs. The norms may be estimated *a priori*, or computed dynamically during the simulations. This approach is generally applicable to adaptive wavelet methods on staggered grids.

One of the strengths of the adaptive wavelet method is that the criteria for adapting (or coarsening) the grid are quantitative and well-defined (13). These criteria also explicitly control the reconstruction error of each variable, and implicitly the tendency error. This contrasts with AMR, where the grid refinement criteria are more qualitative and less systematic. AMR grid refinement depends on a more or less *ad hoc* criterion based on only a subset of the variables (often only the vorticity or velocity gradient). It is therefore not always clear that the AMR approach properly controls the approximation error for all variables, and control of the tendency error is not guaranteed. For example, Ref. [23] experimented with various AMR refinement criteria, and concluded that for a simple shallow water simulation on the sphere

> All three test cases demonstrated that a variety of AMR criteria and thresholds lead to improvements in the results, though to maximize that improvement, the refinement criteria needed careful tailoring.

while they conjecture that for more complex PDEs

> ... more advanced criteria than just a simple relative-vorticity threshold need to be investigated. They could be based, for example, on combinations of physics-based properties (like rainfall), thresholds of vorticity, or gradients. Future work will explore such refinement criteria in the 3D nonhydrostatic version of the Chombo-AMR model ...

An objective grid refinement criterion makes adaptive wavelet techniques a good choice for more complicated multi-physics models.

### 4.4. Extension to Global Models on the Sphere

On the sphere the primal (triangle) grid shown in Figure 1 is generated first, by repeated bisection starting from the icosahedron. Then the dual (hexagon grid) is generated from the primal grid using the perpendicular edge bisectors of each triangle edge as the hexagon edges. A coarsest level is chosen (typically $J_{\min} = 5$ or $J_{\min} = 6$ bisections of the icosahedron). This coarsest level is then optimized to improve its regularity and numerical properties [62]. A multiscale hierarchy of grids is then constructed by simple edge bisection

of the primal grid, starting with the coarsest (optimized) grid. Although the quality of the grid declines with the number of bisections, we have found good results with five or more refinements (see Figure 2).

On the plane, both the hexagons and triangles are nested and regular. In contrast, on the sphere the hexagons are not nested, and the primal and dual grids are not regular. Because the hexagons are not nested, the overlapping regions between fine and coarse hexagons must be carefully accounted for when computing flux restrictions (recall that fluxes are defined across hexagon edges). Grid irregularity means that geometric information must be stored for each (active) cell, and that the 12 pentagonal cells must be treated specially. An example sequence of nested primal grids on the sphere is shown in Figure 4. The resulting wavelet transform is based on the spherical biorthogonal wavelet transform introduced in [59]. Note that all TRISK computations use spherical geometry (e.g., spherical cap areas).

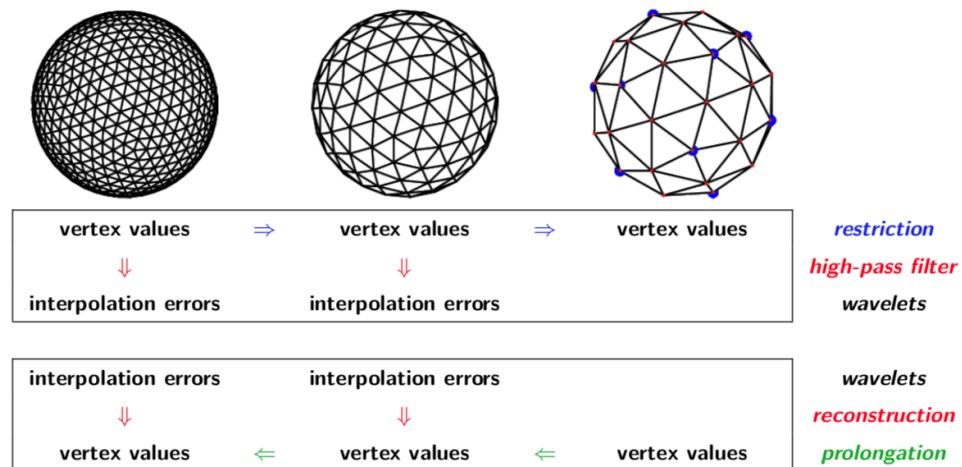

**Figure 4.** Wavelet transform on a non-adaptive primal grid with three scales. (Reproduced from [33]).

### 4.5. Extension to Three Dimensions

Three-dimensional atmosphere and ocean models are constructed from two-dimensional layers (typically 20 to 80 layers). The adapted three-dimensional grid is computed by applying the algorithm described in Section 4.3 to each vertical layer separately, and then taking the union of adapted grids over all layers. This produces a data structure of columns of variable horizontal size, where the width of each column is set by the strictest criterion over all layers.

Implementing only horizontal adaptivity greatly simplifies the code, as well as being consistent with the primarily two-dimensional dynamics of ocean and atmosphere flow. A data structure of columns is also ideally suited to existing parameterization schemes (e.g., for vertical diffusion or solar flux), which expect to work on columns. The disadvantage is that two-dimensional adaptivity is not optimal for tilted structures, or intrinsically three-dimensional phenomena like convection (although our current hydrostatic and Boussinesq model does not support convection, except through parameterization).

Working with columns is also advantageous for parallel load balancing, since the columns can be easily distributed amongst the available cores. Adding more vertical levels therefore improves efficiency since it adds work to each core.

We use a Lagrangian vertical coordinate in both the atmosphere and ocean models. This has the advantage that we do not need to compute vertical fluxes (and vertical velocity is not a prognostic variable). However, in order to avoid layer collapse we need to remap the vertical grid periodically, using a conservative remapping scheme (e.g., a piecewise parabolic method [63]).

In principle, remapping could be used to provide a kind of r-refinement by optimizing the location of the vertical layers to minimize errors at each remapping (e.g., isopycnal layers in an ocean model). It could even be possible to take advantage of so-called dormant layers to provide vertical adaptivity equivalent to h-refinement. The idea is to locally deactivate or reactivate some vertical layers based on an error criterion. (Note that the vertical adaptivity would not necessarily be wavelet-based.)

Figure 5 illustrates the performance of the three-dimensional WAVETRISK atmosphere model with results from the Held & Suarez three-dimensional general circulation test case [64]. This test case uses simplified "physics" (i.e., radiation and friction/drag models) that produce realistic general circulation over relatively short time scales of $O(100)$ days.

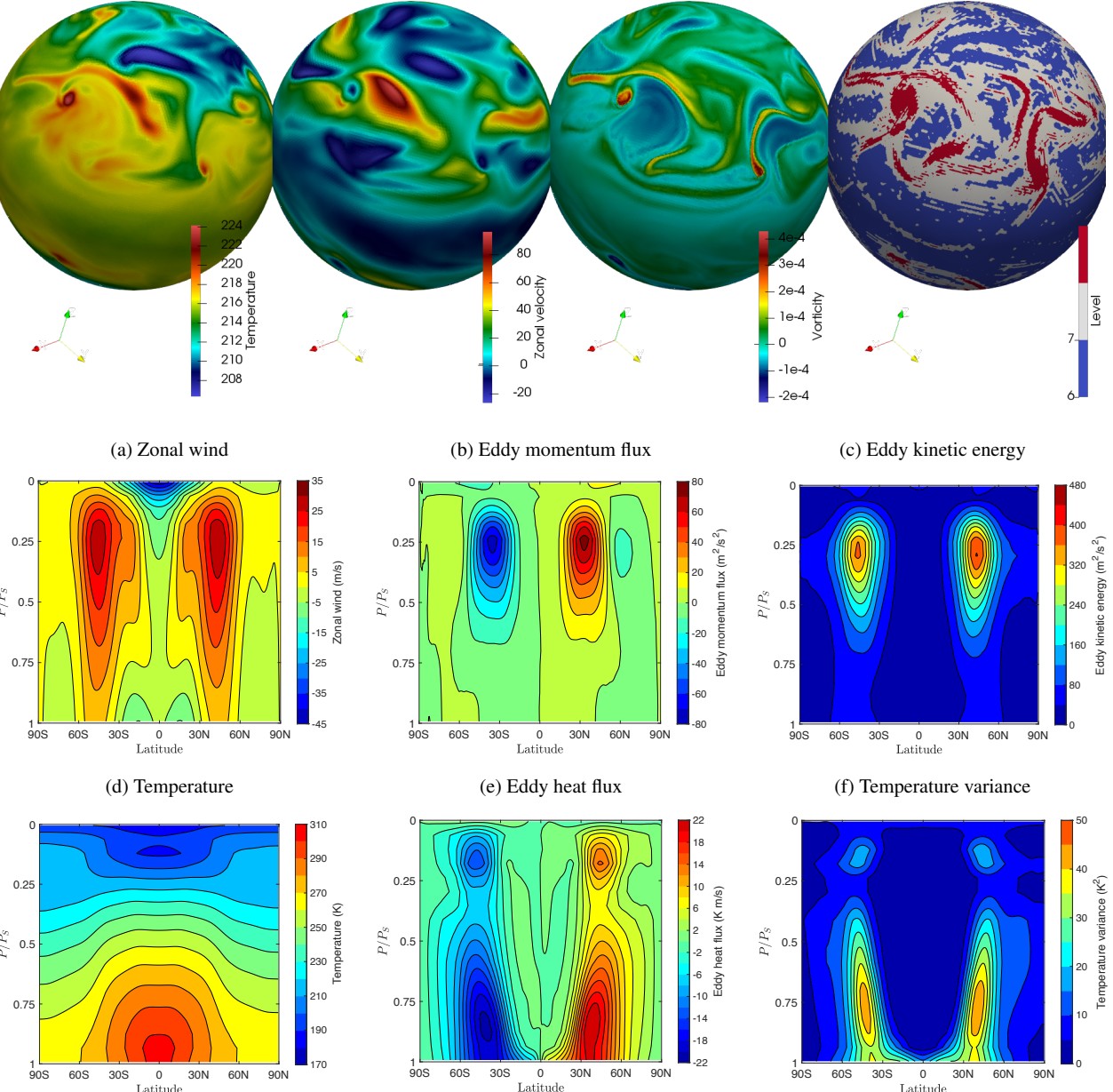

**Figure 5.** Results for the Held and Suarez general circulation test case at 250 hPa. The grid is adapted on the solution with relative error tolerance $\tilde{\varepsilon} = 0.02$. The top row shows the solution at the height 250 hPa, and the bottom rows show zonally averaged vertical slices of important physical quantities. (Adapted from [33].)

### 4.6. Parallelization

Adaptive methods are designed for large problems, and must therefore take advantage of parallel computation. WAVETRISK is parallelized using `mpi` and uses load balancing to redistribute the computational load over the cores as the adaptive grid changes [31]. The data structure is based on the 20 lozenges of the icosahedron, subdivided to the level of the coarsest grid. These $20 \times 4^{J_{\min}}$ lozenges are then distributed the the cores based on their computational load. In order to improve efficiency, the finest data element is a $4 \times 4$ or $8 \times 8$ patch, which includes all grid points (active and inactive). (Patches with no active grid points are, of course, deleted from the data structure.) This produces a so-called hybrid data structure that can be optimized for computational performance by modifying the the coarsest scale $J_{\min}$ and the patch size.

Even though it has not been extensively optimized, WAVETRISK nevertheless shows good strong parallel scaling, as illustrated in Figure 6. Full details of the hybrid data structure and parallelization strategy is given in [31].

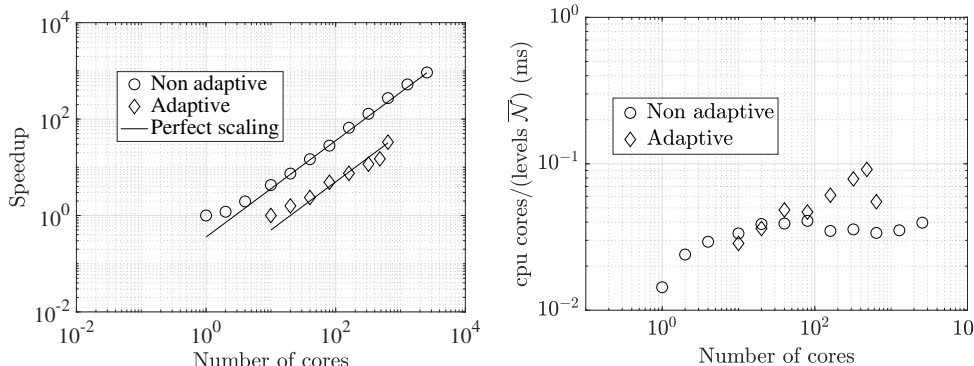

**Figure 6.** Strong scaling of `wavetrisk` for a simulation of the [64] general circulation experiment for a perfectly balanced (non-adaptive) run at a resolution of $J_{\min} = 8$ ($1/4°$) and for a strongly unbalanced (dynamically adaptive) run at a maximum resolution of $J_{\max} = 9$ ($1/8°$) resolution with trend based error tolerance $\tilde{\varepsilon} = 0.08$. Left: speed-up compared with perfect (linear) scaling. The non-adaptive case has perfect linear scaling for more than 8 cores while the adaptive case has power law scaling of approximately 0.78. Right: absolute strong scaling performance in milliseconds (ms) (wall-clock time per time step multiplied by the number of cores, i.e., cpu hours per time step, divided by the average number of active nodes over all vertical levels and all scales). $\overline{\mathcal{N}}$ is the average number of active nodes over all scales. (Reproduced from [33]).

## 5. Local Time Stepping and Multigrid Method for Elliptic Equations

The preceding section gave a detailed example of how an adaptive wavelet method can be developed for semi-realistic atmosphere and ocean models. In this section we describe two additional techniques that take advantage of the resulting adaptive multiscale grid structure.

Currently WAVETRISK uses a single time step based on the finest grid scale on the adaptive grid at any given time. If only a small proportion of the total number of cells are at the finest grid scale this is clearly not an optimal approach. In this case it is much more efficient to use a scale-dependent time scale. In local time stepping the time step is proportional to the local grid scale, with larger scales advancing faster than smaller scales.

Local time stepping was pioneered for adaptive wavelet methods by [65]. They followed earlier work on single stage local time stepping in AMR methods [15]. However, the approach in [65] is limited to second-order in time (i.e., second-order Runge–Kutta). More recently, Ref. [66] extended local time stepping to arbitrary order in time, taking fourth order Runge–Kutta as an example.

Although local time stepping may provide large efficiency gains in some problems, it adds significant computational overhead and complexity to the code. In addition, because in two-dimensional grid refinement the number of cells increases like the square of

the scale $j$, there is often a significant number of cells at the smallest scales, even for flows that produce high grid compression ratios. With three-dimensional grid refinement the number of cells increases like the cube of the scale $j$, so that it is even less likely that local time stepping will produce significant efficiency gains. For example, Ref. [65] found that local time stepping produced a speed up of only 15% when implemented in an adaptive wavelet simulation of combustion.

Geophysical codes often require elliptic solvers, for example to enforce the divergence-free condition, or for implicit time stepping (an implicit free surface barotropic–baroclinic mode splitting method is used in WAVETRISK-OCEAN). The multiscale grids generated by the adaptive wavelet method provide the foundation for an adaptive multigrid solver, at no additional cost.

The standard multigrid solver [67] accelerates the solution of elliptic equations by decomposing a uniform Cartesian computational grid into a sequence of nested grids. The finest grid is the original grid and the coarsest grid is typically $2 \times 2$ (in two dimensions). The highest frequency errors are damped on the finest grid using a few Jacobi or Gauss–Seidel iterations. This approximate solution is then restricted to the next coarsest grid, where the lower frequency errors are damped. This process of restriction and damping is continued until the coarsest grid is reached. The elliptic problem is then solved to machine precision on the coarsest grid (e.g., using bicgstab). The process is then reversed, using appropriate prolongation operators, to produce a so-called V-cycle. Elliptic problems typically converge in two or three V-cycles.

We proposed an adaptive wavelet collocation multigrid solver in flat topology in [68], and later extended this approach to the sphere [69]. The wavelet elliptic solvers use exactly the same V-cycle approach as in standard multigrid, but take advantage of the wavelet prolongation and restriction operators on the adapted multiscale grid to further accelerate the solution.

## 6. Future Directions and Open Challenges

The principal open challenge for realistic dynamically adaptive atmosphere and ocean models is how to incorporate SGS parameterizations. This challenge applies equally to wavelet and AMR approaches, and even to non-adaptive weather and climate models when their resolution is increased. If the physical process is always fully parameterized (e.g., cloud microphysics), then the problem can be solved by using a so-called scale aware parameterization. In this case, the parameterization knows the local grid resolution and adjusts the model appropriately. The more severe case is where the grid has been refined so much that a physical process that had been entirely parameterized becomes (partially) resolved. In this case the dynamical equations and SGS model must both be modified, and the parameterization treats only part of the physics.

In fact, solutions to this problem have already been proposed for the relatively simple case of LES models of turbulence. In this case, only part of the inertial range is resolved and the rest must be modelled. The stochastic coherent adaptive large eddy simulation (SCALES) [70] introduced an adaptive wavelet-based LES model, where the cut-off (filter) scale is adjusted in time and space to resolve the same proportion of the TKE. SCALES provides an example of how to develop SGS parameterizations for other physics. A similar approach for LES of the atmospheric boundary layer was proposed in [28].

We are currently finalizing WAVETRISK-OCEAN, a three-dimensional incompressible hydrostatic ocean model based on the Boussinesq approximation. Since the SGS parameterizations are much simpler in the ocean, WAVETRISK-OCEAN would be a good model with which to explore scale aware parameterizations. For example, the vertical diffusion of temperature and momentum is parameterized using a simple eddy viscosity type TKE closure. Since this parameterization is applied to vertical columns, it should be relatively easy to design a scale-aware parameterization that includes the local horizontal scale. Similarly, Ref. [71] proposed a scale-aware SGS model for horizontal quasi-geostrophic turbulence.

Other approaches to this problem include heterogeneous multiscale modelling (HMM) techniques [72]. In geophysical models HMM has been implemented for super-parameterization of clouds [73], deep convection [74] and surface turbulent mixing [75] in the ocean.

Another future area of research is dynamically adaptive techniques for data assimilation. Data assimilation is essential for accurate weather forecasts [76]. One of its limitations is that the computational grid is not optimized for assimilating the available data, which is usually distributed very inhomogeneously in time and space. The idea would be to dynamically adapt the computational grid to minimize data assimilation error, in addition to accurately compute the dynamics.

## 7. Conclusions

This review has outlined the essential and desirable features a dynamically adaptive wavelet method must have in order to be practically useful for ocean and atmosphere modelling. Foremost among these are discrete conservation of mass and preservation of various important mimetic properties. In addition, the adaptive wavelet method must be compatible with a flux-based, staggered grid discretization. Ideally, it should also use data structures that are typical of geophysical models (e.g., vertical columns, a layer structure). Finally, a global model must use spherical topology, which imposes an irregular grid and particular wavelet transform design.

These features add significantly to the complexity of the code compared to a typical engineering-type adaptive code, but the increased development time leads to big gains in efficiency and accuracy. We illustrated these issues by reviewing how our dynamically adaptive wavelet code for three-dimensional atmosphere and ocean modelling, WAVETRISK, was designed to meet these challenges. We hope that this example will inspire others to build adaptive wavelet methods for geophysical flows.

Although we have focused primarily on wavelet-based methods, many of the considerations we discuss here are also relevant for AMR. AMR techniques for ocean and atmosphere modelling remain an active area of research, and there is not yet an operational three-dimensional global AMR model of the ocean or atmosphere. Progress in adaptive modelling will benefit both the AMR and wavelet communities.

Finally, we reviewed some of the outstanding challenges for adaptive geophysical models. The biggest challenge is how to modify the SGS parameterizations of physical processes so they are compatible with dynamically changing local grid resolution. Progress on this problem will also benefit current operational non-adaptive weather and climate models, which must be extensively re-tuned each time the grid resolution increases. Operational models also make extensive use of nested grids (i.e., static refinement).

The potential of dynamically adaptive techniques to improve realistic geophysical models remains largely unexplored. The only way to know how much they can contribute is to try to actually build them, and then make them as realistic as we can!

**Funding:** This research was funded by an NSERC Discovery Grant.

**Data Availability Statement:** Please see cited articles.

**Acknowledgments:** The author thanks Matthias Aechtner, Jahrul Alam, Laurent Debreu, Thomas Dubos, Marie Farge, Dan Goldstein, Florian Lemarié, Mani Mehra, Kai Schneider, Giulano De Stefano and Oleg Vasilyev, amongst many others, for contributing to various parts of the work described here.

**Conflicts of Interest:** The author declares no conflict of interest.

## Abbreviations

The following abbreviations are used in this manuscript:

| | |
|---|---|
| AMR | Adaptive Mesh Resolution |
| HMM | heterogeneous multiscale modelling |
| LES | Large Eddy Simulation |
| MPAS | Model for Prediction Across Scales |
| PDE | Partial Differential Equation |
| SCALES | Stochastic Coherent Adaptive Large Eddy Simulation |
| SGS | subgrid scale |
| TKE | Turbulent Kinetic Energy |

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
