# Peer review of "Adaptive Wavelet Methods for Earth Systems Modelling"

_fluids, doi:10.3390/fluids6070236_

Round 1

Reviewer 1 Report

Please see PDF

Author Response

Responses to Reviewer 1

Comment: The paper entitled Adaptive Wavelet Methods for Earth Systems Modelling reviews the wavelet adaptive method for geophysical flows. As a reviews it does not contain new material but nicely summarizes the core challenges, the state of the art and the open problems. It is interesting and well written. I could follow the paper very well, but for a deep understanding the cited literature needs to be consulted, which I did not do in detail. However, it serves a very good introduction for people from the geophysical and more technical fluid mechanics community.

Kind regards!

Response: thank you for your review and for picking up on the typos below. I’m glad you found the paper interesting. All changes are indicated by blue text in the revised manuscript.

Comments: Some links are broken, eg ref [33] https://doi.org/doi.org/10.5194/gmd-12-4901-2019

please check all

Response: thanks for picking up on this.  I was using web of science’s bibtex records export feature, but it seems that they are not compatible with the bibliography style used here.  I have carefully checked and corrected all doi links.

Comment: I find the use of ? and ? confusing. Consider adding a subscript

Response: I have used a tilde for the wavelet thresholding parameter to make the difference clearer.

Comment: l.315 is ? a lower bound of ?, i.e. ?≤?? If so please add. Consider using ???? or ????

Response: I have made these changes throughout.

Comment:

• l 372 no indenting

• l 374 no indenting

Response: The Fluids template requires spaces after all equations for correct line numbering in the review stage.  The extra indents will be removed in the final version.

Comment: ref 22 Link does not seem to work

Response: corrected.

Reviewer 2 Report

This paper reviews the use of wavelet-based techniques to design adaptive resolution algorithms for geophysical flows. In the author's 'WAVETRISK' scheme, a wavelet approach is used to adaptively refine (i.e. in both space and time) an initially uniform, structured grid; introducing additional mesh resolution to mitigate numerical errors, typically adjacent to large gradients / filamentary flow features. The so-called TRiSK (Thuburn, Ringler, Skamarock, Klemp, JCP, 2009) mimetic finite-difference/volume discretisation forms the core of the WAVETRISK discretisation, enabling use of general Voronoi-Delaunay meshes.

A number of interesting aspects of the wavelet-based approach are reviewed, including: a volume penalisation type treatment of coastlines/topography, design of restriction/prolongation operators, design of grid adaptivity thresholds, notes on parallel implementations, and possible extensions to incorporate local time-stepping schemes and multigrid solvers. These discussions provide a nice, if yet brief, introduction to the methods in question.

Overall, I feel that a good overview of the mathematical/algorithmic aspects of the wavelet-based methods are presented, but would encourage additional discussion, and perhaps enumeration of challenges, regarding extensions of these methods to full Earth System simulation:

- A number of the flows presented focus on shorter time-scale configurations that contain isolated sharp travelling fronts (i.e. tsunami case). While the performance of the wavelet-based scheme is very impressive in such configurations, it's not clear that full scale geophysical flows are possessed of the same type of localised physics, e.g. global oceanic flows are typified by mesoscale eddies distributed over a large fraction of the domain, while global atmospheric flows feature filamentary vortex dynamics throughout. The characteristic timescales involved also differ significantly; tsunami-type dynamics evolve over a period of hours, while global climate physics evolve over multi-year scales. Are the wavelet-based methods designed with global climate scale configurations as a target?

- Variable resolution (unstructured) mesh configurations are, in some sense, an alternative to the wavelet-based techniques described here, and are used in, for example, various MPAS-Ocean / MPAS-Atmosphere runs to (statically) resolve local-scale dynamics. The dynamic refinement capabilities of the wavelet approach seem to be a clear advantage compared to static, unstructured meshing methods, but are other considerations of computational time/space/efficiency/accuracy important to consider? Does the wavelet-based approach offer significant improvements compared to unstructured variable resolution MPAS-style simulations?

- The discrete energy conservation properties of the TRiSK discretisation is thought to be a major factor in maintaining balanced geophysical flows over long timescales, do the restriction/prolongation operators introduced here conserve energetics? While (potential) vorticity does satisfy a physical conservation law, the TRiSK scheme doesn't in general conserve PV. The design of mimetic restriction/prolongation operators appears to be rather non-trivial, but perhaps additional discussion on the importance of maintaining certain mimetic properties could be included? 

- line 103: repeated word 'method'.

Author Response

Responses to Reviewer 2

Comment: This paper reviews the use of wavelet-based techniques to design adaptive resolution algorithms for geophysical flows. In the author's 'WAVETRISK' scheme, a wavelet approach is used to adaptively refine (i.e. in both space and time) an initially uniform, structured grid; introducing additional mesh resolution to mitigate numerical errors, typically adjacent to large gradients / filamentary flow features. The so-called TRiSK (Thuburn, Ringler, Skamarock, Klemp, JCP, 2009) mimetic finite-difference/volume discretisation forms the core of the WAVETRISK discretisation, enabling use of general Voronoi-Delaunay meshes.

A number of interesting aspects of the wavelet-based approach are reviewed, including: a volume penalisation type treatment of coastlines/topography, design of restriction/prolongation operators, design of grid adaptivity thresholds, notes on parallel implementations, and possible extensions to incorporate local time-stepping schemes and multigrid solvers. These discussions provide a nice, if yet brief, introduction to the methods in question.

Overall, I feel that a good overview of the mathematical/algorithmic aspects of the wavelet-based methods are presented, but would encourage additional discussion, and perhaps enumeration of challenges, regarding extensions of these methods to full Earth System simulation:

Response: thank you for your careful review of the manuscript and helpful suggestions.  I have responded to your specific comments below. Note also that I outline some of main remaining challenges in Section 6.  Changes to the manuscript are indicated in blue in the revised version.

Comment: A number of the flows presented focus on shorter time-scale configurations that contain isolated sharp travelling fronts (i.e. tsunami case). While the performance of the wavelet-based scheme is very impressive in such configurations, it's not clear that full scale geophysical flows are possessed of the same type of localised physics, e.g. global oceanic flows are typified by mesoscale eddies distributed over a large fraction of the domain, while global atmospheric flows feature filamentary vortex dynamics throughout. The characteristic timescales involved also differ significantly; tsunami-type dynamics evolve over a period of hours, while global climate physics evolve over multi-year scales. Are the wavelet-based methods designed with global climate scale configurations as a target?

Response: The wavelet-based methods are designed modelling over short (days), medium (years), and climate scaling modelling (decades).  The necessary property is that the flow compresses reasonably well in the wavelet basis. As shown by the Held and Suarez (1994) simplified climate test case in Figure 5, adaptivity can provide significant grid compression and accurate results for climate type simulations.  In Aechtner, Kevlahan & Dubos (2014) Figure 17 we showed that even for the “worst” case of homogeneous isotropic two-dimensional turbulence on the sphere, wavelets provide a significant reduction in the the number of grid points compared to a non-adaptive method.  We are currently evaluating WAVETRISK-OCEAN on a baroclinic jet instability test case (Soufflet et al Ocean Modelling 98, 2016) that runs for several years.   One advantage of the adaptive variable resolution framework of the wavelet methods is that it is easy to do a climate run at relatively low resolution for a long period to spin up and reach steady state, and then restart with a higher resolution for shorter periods. As I’ve mentioned in the paper, we are still at the beginning of developing and evaluating any type of adaptive Earth System models.  The question of the usefulness of AMR or other adaptive methods remains open, but they show promise.

Comment: Variable resolution (unstructured) mesh configurations are, in some sense, an alternative to the wavelet-based techniques described here, and are used in, for example, various MPAS-Ocean / MPAS-Atmosphere runs to (statically) resolve local-scale dynamics. The dynamic refinement capabilities of the wavelet approach seem to be a clear advantage compared to static, unstructured meshing methods, but are other considerations of computational time/space/efficiency/accuracy important to consider? Does the wavelet-based approach offer significant improvements compared to unstructured variable resolution MPAS-style simulations?

Response: Variable resolution unstructured meshes are used extensively in ocean modelling to provide higher resolution near coastlines, and have been proposed to improve representation of orography (e.g. the Andes mountain range).  However, this approach can produce instabilities and does not always improve accuracy. A multiscale structure of regular grids may have better numerical properties than a single scale unstructured mesh where the geometrical properties are not easy to control (e.g. triangles can have extremely acute angles). Unstructured mesh generation is an open area of research, especially in aeronautics. On the other hand, the multiscale scale structure of the wavelet method adds significant additional computational overhead (about a factor of two for each active grid point).  This means that a grid compression factor of more than two is necessary for the adaptive wavelet method to be more efficient.  I have added a discussion on these points on page 2, as well as more details on grid nesting, including a new reference to a paper by Santos and Peixoto where they evaluate variable resolution for the Andes mountain range.

Comment: The discrete energy conservation properties of the TRiSK discretisation is thought to be a major factor in maintaining balanced geophysical flows over long timescales, do the restriction/prolongation operators introduced here conserve energetics? While (potential) vorticity does satisfy a physical conservation law, the TRiSK scheme doesn't in general conserve PV. The design of mimetic restriction/prolongation operators appears to be rather non-trivial, but perhaps additional discussion on the importance of maintaining certain mimetic properties could be included? 

Response: There are two variants of TRiSK: one that conserves total kinetic energy and another that conserves total potential enstrophy. Both variants conserve potential vorticity, both locally and globally:

“Compatibility in the sense defined here is sufficient to guarantee local and global conservation since the sole forcing term in the discrete PV equation is the divergence of a flux.” (Ringler et al 2009)

The restriction and prolongation operators do conserve energy (as well as the other mimetic properties of TRiSK).  However, a small amount of energy is lost during the grid adaptation step (most is conserved during interpolation onto the new adaptive grid).  This energy is equivalent to an extra horizontal diffusion.  In practice, most climate models include a small amount of diffusion, using a Laplacian or bi-Laplacian operator, to damp grid scale noise.  I’ve added some more discussion on pages 8 and  10 on energy conservation and other mimetic properties. 

Comment: line 103: repeated word 'method'. 

Response: Corrected.